# Spatial–Temporal Variation and Influencing Factors on Housing Prices of Resource-Based City: A Case Study of Xuzhou, China

Qing Yao  and Yingen Hu *

School of Public Management, Huazhong Agriculture University, Wuhan 430070, China;
2019315010014@webmail.hzau.edu.cn
* Correspondence: hyg@mail.hzau.edu.cn

**Abstract:** The steady development of the real estate market is an important link in the transformation of resource-based cities. Taking the main urban area of Xuzhou, a typical resource-based city in China, as an example, this study used a spatial–temporal geographically weighted regression (GTWR) method guided by characteristic price theory to analyze the evolution of the spatial–temporal pattern of house prices in resource-based cities and the influencing factors. The results of the study showed that, from the time trend, Xuzhou's main city house price trend underwent the obvious stages; from June 2015 to September 2016, the trend was stable, and from March 2017 to June 2018, it was an "up–down–up" trend. With respect to the spatial distribution of house prices, from June 2015 to June 2017, the traditional business district and the major city in the Xuzhou metropolitan area were the areas of highest house prices, and after September 2017, the structural features of the traditional business district, along with the main city of the metropolitan area and the new city, formed the areas of highest importance. With regard to the factors influencing house prices, the type of dwelling, according to the building characteristics, had the largest impact on house prices, and the enrichment of housing types became an effective way to regulate housing prices. The impact of location characteristics on house prices varied depending on differences in the public infrastructure surrounding housing, with the contribution of the planned metro stations to house prices not effectively emerging. The impact of neighborhood characteristics on house prices varied, with tertiary care hospitals having a 'neighborhood avoidance effect' on house prices, given that hospital-generated waste was too densely distributed around houses, suppressing neighborhood house prices. The results of this study indicated that, in the process of real estate market development in resource-based cities, the planning department should consider the different functions and division of labor in each region and scientifically formulate urban development plans to provide a good external environment for the healthy development of the city's property market.

**Keywords:** resource-based city; housing prices; GTWR model; spatial pattern evolution; influencing factors

## 1. Introduction

A resource-based city is a special type of city that has emerged as a result of resource extraction and relies on resource industries to support its development [1]. The 14th Five-Year Implementation Plan clearly proposed that promoting the high-quality development of resource-based areas is an important step in overcoming the shortcomings of transformation and development. The healthy development of the real estate market is an important part of the transformation and green development of resource-based cities and is directly related to the effectiveness of high–quality development. There are currently 262 resource-based cities in China, nearly half of the total number of prefecture-level cities in the country [2]. The problem of falling house prices caused by population exodus and hollowing out of local areas in the transformation of resource-based cities is more alarming [3] in the context of differentiated real estate control policies such as 'city-by-city' and 'one-city–one-policy' [4].

The responsibility for real estate control has been placed more on local government, so further improving policy relevance and providing housing security for residents has become an urgent issue for the local government. Therefore, determining the characteristics of the evolution of the temporal and spatial patterns of residential prices in resource-based cities and further clarifying the main factors influencing their house prices can provide a basis for the scientific formulation of real estate regulation and control policies.

At the end of the 1990s, the central government carried out macro-control on the real estate industry for the first time. Through continuous exploration and practice, it gradually defined the characteristics of regulation based on administrative means [5]. "Government failure" and "market failure" coexist, so the long-term regulation mechanism of the real estate market was not well established. In 2005, the central government issued the Notice on Effectively Stabilizing Housing Prices, proposing eight measures to curb excessive housing price rises (commonly known as the Eight Measures of the New Country). In 2010, the Notice of the State Council on Resolutely Curbing the Excessive Rise of Housing Prices in Some Cities (known as the most stringent regulatory policy in history) and other documents repeatedly emphasized the suppression of excessive housing price increases [6]. Judging from the market situation over the years, it seems that they have not achieved the expected results [7]. It was not until 2014 that China changed its usual national "one game of chess" and "one size fits all" regulatory policy and instead implemented a "categorical regulation" and "policy based on each city" regulatory policy [4]. Recently, the government work report of the 19th National Congress of the Communist Party of China clearly pointed out that we must adhere to the position of "housing is for living in, not for speculation". That highlighted the great importance the central government attaches to the regulation of the real estate market [8]. Therefore, scientifically examining the changing laws of regional real estate is an important issue related to the healthy development of China's real estate. As a typical resource-based city, Xuzhou has made significant achievements in industrial transformation and the utilization of renewable resources in recent years and won the 2018 United Nations Habitat Award [9]. By December 2018, the average house price increased from CNY 5458 per square meter to 10,132 CNY per square meter (data source: Anjuke.com). The rate of increase in housing prices was characteristic of the current period of economic development and transformation [10]. Exploring the law of property prices in Xuzhou is thus a key issue for the management of the local property market. This is a pressing problem that needs to be addressed, and the premise for addressing this problem is to scientifically examine temporal and spatial changes and influencing factors in housing prices.

## 2. Literature Review and Theoretical Framework

### 2.1. Hedonic Price Theory

Hedonic price can be understood as meaning "to be able to bring usefulness to people". In the hedonic price model, commodities are thought to be composed of many different characteristic attributes. And the price of a good is the sum of certain characteristic prices contained in the good that can satisfy the needs of the population. The various characteristics combine to form a set of characteristics that comprise the total effect on consumers; what consumers actually purchase is a collection of these effects. During the 1960s, Lancaster and others introduced hedonic pricing theory into research in real estate and urban economics, forming the theory of hedonic housing prices. The characteristic price function relates to the functional relationship between the total price of housing and the characteristic quantity of housing. The law of diminishing marginal utility implies that the upward trend in the product characteristic price curve will gradually slow and the marginal price of the characteristic will fall [11]. Spatial and temporal geographically weighted regression (GTWR), which is based on spatial–temporal nonsmoothness, calculates the spatial–temporal distance of each sample point, expanding the study sample size in the time dimension [12]. Wang et al. applied the method to analyze the factors that influence house prices in cities such as Beijing [13] and Shenzhen [14,15] and verified the high estimation

efficiency as well as the goodness of the model. Placing house prices in a spatiotemporal perspective for year-round analysis allows for an accurate capture of the dynamics of house prices, thereby providing a theoretical benchmark for scientific regulatory and control policy formulation.

### 2.2. Advances in Research on the Spatial Effect of Housing Prices

Geographer Waldo R. Tober proposed the first law of geography in 1969. That is, there is a certain spatial influence relationship between economic behaviors and the closer the distance, the greater the mutual influence may be [16]. In a lay person's terms, it can be said that "everything is related, the closer it is, the more related it is". In a study of the spatial effect of housing prices, the spatial effect of housing prices in the UK was the first to receive attention [17,18]. Oikarinen and Browning first proved the existence of spatial effects in real estate research [19] and introduced the spatial spillover effect into the real estate field. Their research determined that housing prices in a region can affect the changes in housing prices in surrounding areas under the action of element flow.

In recent years, relevant international literature has introduced spatial effects into characteristic price models and used quantile regressions for impact factor analysis [20]. In addition to studies on real estate bubble measures [21–25], other aspects have been gradually advanced. The models based on spatial econometric analysis tend to be advanced and diversified. These models are constructed to quantify real-world problems in detail. Cohen studied the significant spatial spillover effect of residential price growth rates in US cities [26]. Badi used a spatial lag model with nested random effects and included instrumental variables to analyze the "annual round effect" in residential prices in the UK [27]. Beenstock constructed a spatial unbalanced panel model to study the residential price spillover effect [28]. Hyun constructed a spatial autoregressive model to compare and analyze the difference in spatial dependence between boom and recession periods, determining that the spatial dependence characteristics are higher in boom periods than in recession periods [29]. Mathur constructed a two-stage spatial quantile regression model to study the impact of urban expansion boundaries on residential prices, pointing out that there is spatial disequilibrium in the impact of urban expansion boundaries on residential prices [30]. In addition, the geographically weighted, characteristic price model of regression can better reflect the spatial heterogeneity of influencing factors and this model has also been applied to relevant empirical studies [31].

Although studying the spatial effect of residential price fluctuations began late in Chinese research, relevant research results focus on the effects of the spatial spillover of residential prices on the economic fundamentals and the social problems caused by high residential prices [32], using national [33], varying urban clusters [34], provincial [35] or municipal [36], and single-city [37,38] data to examine the spatial effect of residential prices.

### 2.3. Theoretical Analysis

Based on the microlevel and under the guidance of the hedonic price theory, we start with the three aspects of architectural characteristics, location characteristics, and neighborhood characteristics [39,40]. The factors affecting housing prices in resource-based cities were analyzed from the microlevel, without considering the impact of macrolevel and land prices on housing prices. (1) Construction factors included five factors: house age, greening rate, floor area ratio, property fee, and housing type. Generally speaking, the older the age of the house, the greater the depreciation and maintenance costs and thus a negative impact on housing prices. The greening rate refers to the ratio of the sum of various green areas within the land-use area to the total land area. The higher the greening rate, the higher the comfort of the residents, the fresher the living air, and the more expensive the housing price. The residential floor area ratio refers to the ratio of the total construction area to the total land area, also known as the gross density of the construction area [41]. A higher floor area ratio usually indicates that the housing construction is dense, which will correspondingly lead to a high-density residential population, and the

comfort of residents will be lost. The property fee can reflect the level of property service in the community. The property fee in a community with high housing prices is usually higher. However, excessive property fees will increase housing costs or holding costs, so there is uncertainty regarding the impact of property fees on housing prices. Low-rise residential buildings are the most suitable for living [42]. Low-rise residential buildings with lower housing age are usually more expensive. The impact on housing prices cannot be generalized. (2) Location factors include the planned subway line, main road distance, and government distance. Simply put, the presence or absence of a subway has become a symbol of a city's level. Whether a subway can be built depends on both the level of economic development of the city and the size of the city. As the city of Xuzhou expands, it has the qualifications to build a subway. Theoretically speaking, the construction of the subway can improve traffic accessibility, facilitate citizen travel, and promote the rise of housing prices [43]. When buying houses, residents tend to buy near the main road for the convenience of transportation, and the main road brings convenient transportation to the owners; real estate near the main road usually has a higher price. Government residences are usually surrounded by the core areas of government affairs, and these areas will quickly mature and become prosperous as the government moves in. In the areas that are closer to the government, the surrounding houses are more expensive. (3) Neighborhood factors include distances from primary schools, middle schools, and institutions of higher learning, distances from tertiary hospitals, and distances from tourist attractions. Areas close to schools are rich in educational resources and rich in cultural atmosphere, making it convenient for students to enroll nearby; this indicator can promote higher housing prices. Tertiary hospitals, as a scarce medical resource in cities, can promote higher housing prices. Generally speaking, a surrounding environment of tourist attractions is beautiful, and the potential for housing price appreciation is great.

The research concept of this paper is illustrated in Figure 1. First, based on the GIS spatial analysis function, the spatial and temporal distribution of house price data in the study area was visually represented in order to analyze the spatial and temporal variation characteristics of house prices in the study area. Second, based on the characteristic price theory, the potential factors affecting house prices were analyzed in depth from a theoretical perspective. Again, a spatiotemporal, geographically weighted regression model considering spatiotemporal heterogeneity was constructed using multitemporal data, and the results were analyzed in depth; differences in the factors influencing house prices in different residential types were considered, and empirical results for the factors influencing house prices in different types of residential neighborhoods were derived. Finally, based on the empirical results of the study, policy recommendations for regulating house prices are proposed in a targeted manner.

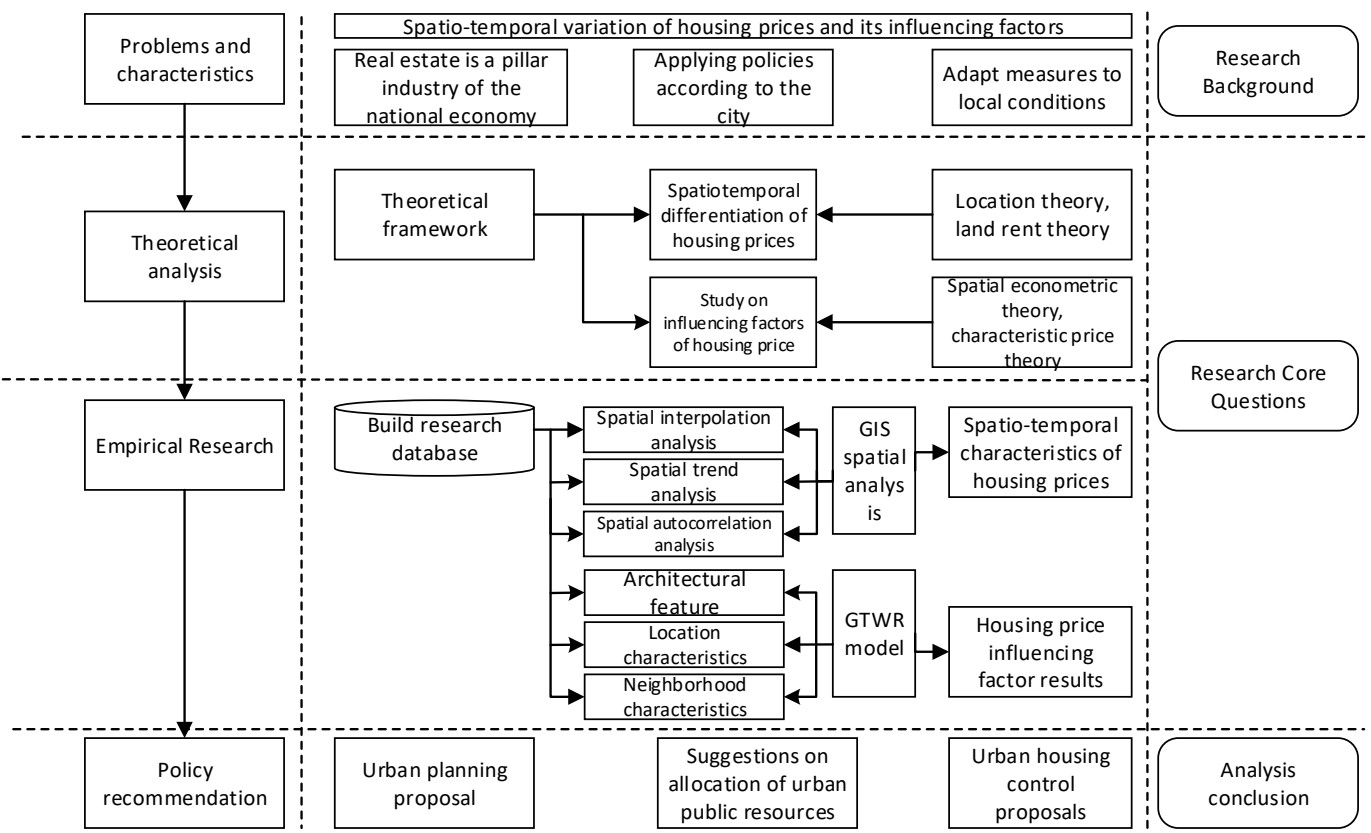

**Figure 1.** Analysis framework.

## 3. Materials and Methods

### 3.1. Study Area

Located in the northwest of Jiangsu Province, Xuzhou is a border city between Jiangsu, Shandong, Henan, and Anhui Provinces and a typical resource-based city. By the end of 2017, the city had a resident population of 8,763,500, an urbanization rate of 63.76%, a GDP of CNY 660,595 million and real estate investment of CNY 54,913 million. The main urban area of Xuzhou includes four municipal districts, Yunlong District, Quanshan District, Gulou District, and Tongshan District, which are the main centers of activity for citizens to live and work in. The traditional business district, the main city of the metropolitan area, and the construction of the new city are all located within the main urban area of Xuzhou. The intersection of Zhongshan North Road and Huaihai Road is a traditional business district within the main city of Xuzhou, where high-class consumer establishments such as Pengcheng Plaza, Golden Eagle Shopping Centre, and Central Baida are concentrated. The area is commercially developed and is the ancient and prosperous business center of Xuzhou. The main urban area is the old urban area of Xuzhou City, which is adjacent to the Yunlong Lake scenic area, with well-developed basic public facilities and well-developed transportation conditions. Construction of the new urban area is located in the southeast of the main urban area, which is the key construction area in the master plan of Xuzhou City, and the area has gradually become the model area of Xuzhou City. Xuzhou City was awarded the United Nations Habitat Award in 2018 and has made outstanding achievements in building a livable city. In terms of development position and development stage characteristics, the total number of major economic indicators in Xuzhou City is close to the national average, and the city's development achievements and challenges are common to the whole country [25]. Taking Xuzhou as the study area, the spatial pattern of residential house prices and the factors influencing it in the process of transformation and development were examined from a micro perspective, which was conducive to further

promoting the healthy development of its real estate market and provided a reference for the regulation and management of house prices in resource-based cities.

The research took 965 communities in the Xuzhou urban area as the research samples, drew on the previous research results [44], and delineated the research scope (as shown in Figure 2. Subsequent interpolation and other analysis in the study area will affect the accuracy of the empirical analysis results. (1) The research area delineated in this paper was based on the distribution of community sample points and basically covered the urban area; the public infrastructure in the research area was sound, the value of regional social services was high, and the research significance was considerable. (2) The distribution of research samples in this area was relatively concentrated, and correspondingly, the data collection of influencing factors was relatively convenient and roughly covered the approximate range of the "one-hour traffic circle" from the city center to the outside [45]. Figure 2 shows the extent of the study area. It can be seen that the distribution of sample points in the study area was relatively uniform (uniform sample distribution can effectively improve the accuracy of interpolation analysis), comprising the densely populated residential area in Xuzhou City.

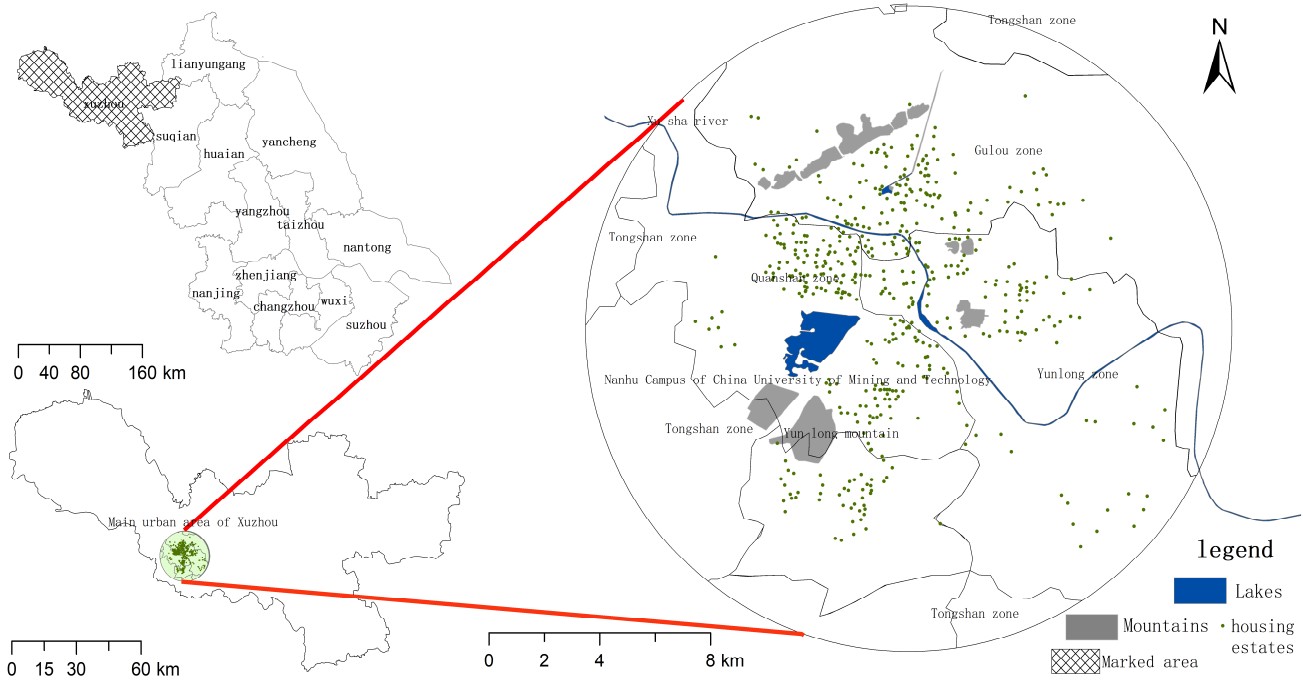

**Figure 2.** Study area and samples data.

### 3.2. Evaluation Indicator

The explanatory variable was the average price of residential commercial properties (in CNY/m$^2$), which aimed to measure the relationship between the price and the size of residential commercial properties. The housing price data used in the research came from the Anjuke website, which is the largest Internet real estate information provider in China (https://xuzhou.anjuke.com, accessed on 9 October 2018). The housing information provided is relatively comprehensive and reliable, and recent research in the field of real estate has mostly used the housing information provided by it. This study used the web scraping method to collect housing price information. The study covers the period from June 2015 to June 2018, using quarterly data published on Anjuke.com, and a total of 965 residential properties in the study area were collected. Considering the comparability of data and eliminating the influence of price level [46], this paper corrected the collected housing price data. The specific method was correcting the housing price data to the research starting year (2015) according to the consumer price level. Since second-hand apartments are considered to be more market-oriented in China and more suitable for

reflecting the personal preferences of urban infrastructure [47], we used the average price of second-hand apartments (in Yuan/m$^2$) of residential communities as the dependent variable. It should be noted that the price of residential commercial properties, as defined herein, refers to the average sales price of commercial properties with a 70-year ownership period and the type of housing being "ordinary apartments".

The definitions and data sources of the explanatory variables are shown in Table 1. Data capture was divided in three ways. First, according to the housing's own characteristic data (house age, residential floor area ratio, greening rate, property fee, and housing type), according to the data published on Anjuke.com. The network collection method was used to collect it, and the missing data were supplemented by visiting local residents on the spot and consulting intermediaries by telephone. Second, the data of points of interest (middle schools, primary schools, tourist attractions) involved in the influencing factors came from the country geographical data sharing platform (http://www.geodata.cn, accessed 4 March 2019). Third, among the other factors, the subway lines and main roads were respectively provided by the author in the "Planning Map of Urban Rail Transit Network in Xuzhou City" (http://www.xzdtjt.com/article/9/10/1854.thtml, accessed on 28 December 2015). The reason Euclidean distance was selected for analysis in this paper is that Euclidean distance describes the relationship between each pixel and a single "source" or "a group of sources" according to the straight-line distance and gives each pixel in the raster to the nearest source distance. The specific operation involved using the "Spatial Analysis" toolbox in the Arc Toolbox analysis tool, clicking to select the "Distance Analysis" function, and realizing the analysis goal according to the "Euclidean Distance" command. The equidistance was calculated from the middle to the planned subway.

**Table 1.** Description statistical analysis of effect variables.

| Characteristic Variables | Explanatory Variables | Data Sources | Calculation Method |
|---|---|---|---|
| Architectural character | Age of room (years) Greenery rate (%) Residential floor area ratio (%) Property Charges (CNY/m$^2$) | https://xuzhou.anjuke.com | Accessed by crawler |
| | Type of dwelling (-) | | 11 stories and above are high rise dwellings assigned a value of 3; 6–11 stories are mid-rise residences assigned a value of 2. Mixed dwellings are assigned a value of 2; low-rise dwellings below 6 stories are assigned a value of 1. |
| Location characteristics | Distance to planned metro stations (km) | Xuzhou Metro Planning Network | Euclidean distance |
| | Distance to main road (km) | Xuzhou City Master Plan (2007–2020) | Euclidean distance |
| | Distance to government (km) | | |
| Neighborhood characteristics | Distance to primary school (km) Distance to secondary school (km) Distance to tertiary hospital (km) Distance to tertiary institutions (km) Distance to tourist attractions (km) | National Geographic Data Sharing Platform | Euclidean distance |

### 3.3. Evaluation Method

3.3.1. Kriging Interpolation

The Kriging interpolation method, also known as spatial local interpolation, is a method that can better handle the interpolation of spatially nonstationary variables. Based

on spatial autocorrelation, this method is an unbiased optimal estimation method for variables in limited areas. It takes into account not only distance relationships, but also the spatial distribution of known sample points in relation to the spatial orientation of unknown sample points through variance functions and structural analysis. In Kriging interpolation, the estimated value of a variable at a point to be estimated is a linear combination of $n$ known variable-valued measurement points in its surrounding sphere of influence, whose mathematical expression is Equation (1) [47]:

$$Z(x_0) = \sum_{i=1}^{n} \lambda_i Z(x_i) \tag{1}$$

where $Z(x_0)$ is the unknown plot house value; $Z(x_i)$ is the value of the known house price around the unknown house price in the region. $\lambda_i$ is the plot house price weight, and $n$ is the number of plot observation points. Since the expected value of house prices in the study was unknown, the ordinary Kriging interpolation method of the spherical function was chosen to detect the interpolation results of house prices in the Xuzhou urban area at each point in time, and since the study area was not large and the sample size was sufficient, the use of ordinary Kriging interpolation met the requirements of the analysis of the evolution characteristics of the spatial–temporal dynamic pattern of house prices.

### 3.3.2. Spatial Trend Analysis

Trend analysis can analyze the global trend of sampling data sets from different perspectives. Each vertical bar in the trend analysis graph given by ArcGIS 10.2 software represents the value and position of the data, and a best fitting curve is made by projection and used to simulate. There is a trend in the data in a specific direction. The more curved the curve, the greater the difference in the data. On the other hand, the spatial differentiation law is not obvious. The specific orientation can be determined according to the coordinate axes, where the $x$-axis represents the east and the $y$-axis represents the west [48].

### 3.3.3. Spatial Autocorrelation Method

Spatial exploratory analysis is the premise of spatial econometric analysis. Only after spatial exploratory analysis determines that the dependent variable has a spatial agglomeration effect can there be a reason to introduce a spatial econometric model. There are many spatial exploratory analysis tools, and the most widely used one is the Moran index. Using the Moran index to measure the spatial correlation of the dependent variable, the expression is as follows [49]:

$$I = \frac{\sum_{i=1}^{n} \sum_{j=1}^{n} W_{ij}(x_i - \bar{x})(x_j - \bar{x})}{\sigma^2 \sum_{i=1}^{n} \sum_{j=1}^{n} W_{ij}} \tag{2}$$

where $x_i$ is the $i$ explanatory variable for the first region, i.e., house value; $n$ is the number of regions; and $\sigma^2$ is the sample variance. $W_{ij}$ is the spatial weight matrix, which describes the mutual adjacency of each evaluation unit in space. If the significance is positive (negative), it indicates that there is a positive (negative) spatial autocorrelation of the level of rural green development. Global spatial autocorrelation assumes spatial homogeneity and cannot reflect the specific agglomeration characteristics and their significance within the study area. Local spatial autocorrelation solves this problem. To further analyze the local spatial clustering and dispersion characteristics of rural green development, local spatial autocorrelation is used for analysis, and local spatial autocorrelation is characterized using a local representation, calculated as [48,49]:

$$I_i = \frac{(x_i - x)}{\sigma^2} \sum_{j=1}^{n} W_{ij}(x_j - \bar{x}) \tag{3}$$

where $x_i$, $x_j$ are the attribute values of evaluation units $i$ and $j$ ($i \neq j$), $\bar{x}$ is the mean value of evaluation unit attributes, $n$ is the total number of evaluation units in the study area (pcs), $I_i$ is the evaluation unit local area and is positive (negative) to indicate the proximity of similar (dissimilar) house price areas, with larger absolute values indicating higher proximity.

3.3.4. Spatial–Temporal Geographically Weighted Regression

The traditional least squares regression method estimates the coefficients globally or on average and cannot accurately reflect the spatial heterogeneity of the regression coefficients, while GTWR takes the spatial and temporal distances into account and can effectively tap the local characteristics of the relationship between the dependent and independent variables. The GTWR model, based on the GWR model, considers spatial location and time series coordinates, as a whole, to form three-dimensional coordinates and considers both spatial and temporal influences on the regression coefficients of each explanatory variable. Its mathematical formula is [50]:

$$y_i = \beta_0(u_i, v_i, t_i) + \sum_{k=1}^{d} \beta_k(u_i, v_i, t_i) + \varepsilon_i, i = 1, 2, \ldots, n \tag{4}$$

where $y_i, x_{i1}, x_{i2}, x_{i3}, \ldots, x_{ik}$ is the value of the dependent variable $y$ and the independent variable $x$ at the observed location; $u_i, v_i, t_i$ is the spatial–temporal coordinate of the $i$ sample point, for spatial distances, calculated using the latitude and longitude of the sample point; $t$ is the temporal distance. $\varepsilon_i$ is the random error term. To reduce the influence of data dimension on model estimation, this paper used the full logarithmic form to estimate the GTWR model [51]. In addition, the bandwidth selection has a large impact on the model accuracy. Referring to related studies, the cross-confirmation method was used to determine the model spatial bandwidth and temporal bandwidth [52], and the Gaussian distance decreasing function was used to determine the spatial–temporal full matrix, which avoided the estimation bias caused by the sparse data of neighboring samples at certain sample points [53].

## 4. Evolution of the Spatial and Temporal Pattern of House Prices

### 4.1. Time Evolution

As can be seen from Figure 3, overall, the average house price in Xuzhou's urban areas rose from 6579.06 CNY/m$^2$ in June 2015 to 10,729.26 CNY/m$^2$ in June 2018, and the trend of house price changes can be divided into two phases. Specifically, from June 2015 to September 2016, Xuzhou's house prices were in a "dormant period", with no significant fluctuations during this period. From March 2017 to June 2018, house prices in Xuzhou showed an increasing trend. The main reason for this phenomenon was the nationwide "de-inventory" policy, which changed Xuzhou from a housing resettlement policy to a monetized home purchase subsidy, directly stimulating the real estate market boom. As of May 2017, the city's commercial housing de-conversion cycle dropped to 7.41 months, and the sales area of commercial properties remained above 20% for 15 consecutive months. In addition, the easing of the net population outflow problem and the construction of green cities and other benign urban developments also provided the external conditions for the real estate market boom. From June 2015 to September 2016, house price trends across the region were largely consistent with the overall house price trend. From March 2017 to June 2018, house prices in Tongshan District and Quanshan District were on the rise, with Gulou District seeing the most significant growth from December 2017 to March 2018, driving up overall house prices. The reason for this phenomenon was closely related to the influx of foreign speculative funds, as individual districts within the traditional business district of Gulou were sensitive to foreign funds and were in a "leading" position, while the rest of the district was in a "following" position. As a result of a combination of factors, house prices in Xuzhou reached historic highs in late 2017 and early 2018. Similar to the house price trend, Xuzhou's house price growth rate peaked at the end of December 2017 to the beginning of March 2018 and then quickly fell back, which was closely related to

the real estate management's rapid response to the price limit policy. The introduction of measures in which developers were not allowed to record prices higher than the previous three months' prices directly curbed the excessive rise in house prices.

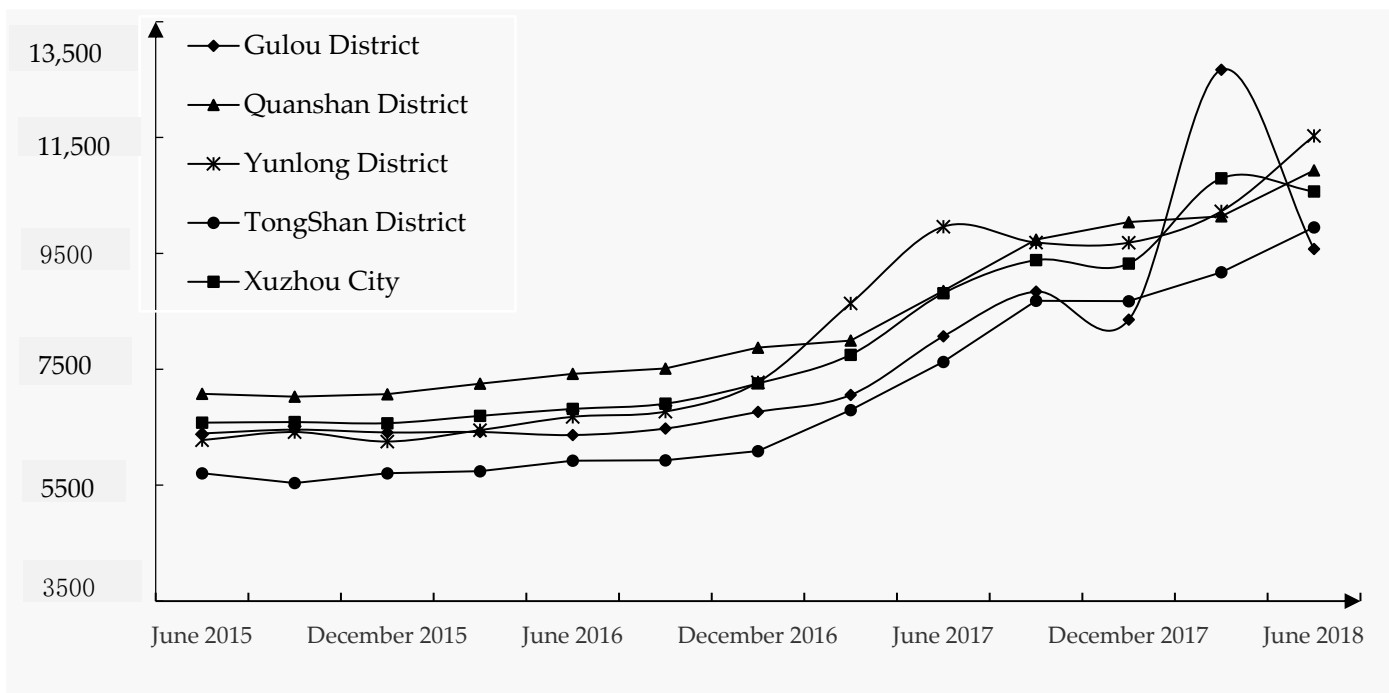

**Figure 3.** Temporal Variation of Housing Price in the Study Area.

### 4.2. Spatial Distribution

According to Equation (1), spatial interpolation of residential prices in the main urban area of Xuzhou for June, September, and December 2015, 2016, and 2017 was carried out using the ordinary Kriging interpolation method. In order to show the evolution of Xuzhou's house price pattern more intuitively, the annual grouping range was divided into five categories: low, low, medium, high, and high using the natural breakpoint method in ArcGIS 10.2 (because of space limitations, only the results of the above time points are shown), and then the distribution of Xuzhou's house price pattern was obtained. The results are shown in Figure 4.

In terms of spatial distribution, house prices in Xuzhou's main urban area in June 2015 roughly showed a decreasing structure from the traditional business district and the main city of the metropolitan area as the peak area for house prices to the periphery. The high-value areas of the study area were mainly located around Yunlong Lake, while the low-value areas were located near the Xusha River. From September 2015 to June 2017, there was a significant expansion in the high-value areas of house prices and a consolidation of the pattern of house price levels between regions. From September 2017 onwards, the high-value areas of house prices were mainly located in the new urban areas of Yunlong construction, and these areas became the new peak areas of house prices because of Xuzhou's "expansion to the southeast" urban development policy [54]. By December 2017, house prices in the new urban areas in the southeast orientation of the study area had surpassed house prices in other areas, such as the main city of the metropolitan area, becoming the peak house price zone. The spatial pattern of the traditional business district to the northwest, the main city of the metropolitan area, and the new urban area to the southeast as a high-value area for house prices had basically taken shape. This was due to the influence of Xuzhou City planning and the high prominence of new urban area house prices. The new city is located in the southeast of the main city, with a beautiful environment and a greening rate of over 36.9%. In total, 1333.33 hm$^2$ of land has been

expropriated, and the full-scale construction of residential development projects has pushed up housing prices.

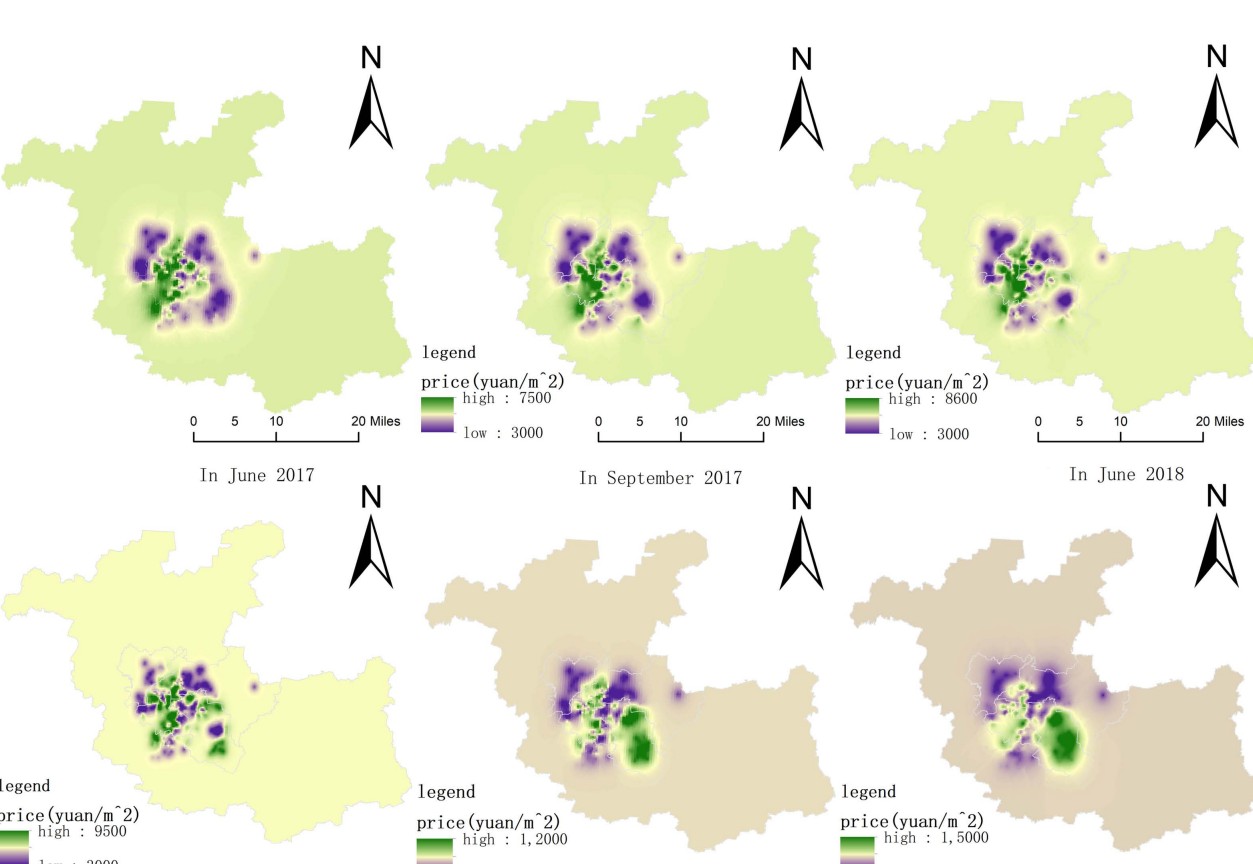

**Figure 4.** Spatial Distribution of Housing Price of Study Area.

### 4.3. Evolution of the Spatial Pattern of House Prices

From Figure 5, it is clear that the spatial pattern of residential prices in the urban area of Xuzhou was relatively stable, and spatial residential price divergence since June 2015 showed a "reverse U-shaped" pattern of divergence from the northwest to the southeast. One possible reason for this result in the analysis of trends is that the urban planning and expansion policy of the city of Xuzhou controlled the pulse of urban development and had a large impact on urban residential prices. Urban expansion means that the "radius of the city" expands, and citizens buy homes in a larger area. Changes in the southeastern part of the city of Xuzhou were evident to all, and the development of the new town was linked not only to industrial transformation but also to economic development and the gathering of talent; the expansion of the residential map greatly boosted the development of its property industry.

### 4.4. Analysis of Spatial Autocorrelation Results

Spatial autocorrelation was evident at the 5% confidence level of housing prices in urban residential neighborhoods in Xuzhou, providing evidence that housing prices had important spatial effects (Table 2). The global Moran and local Moran indices were both positive, indicating that there was a positive spillover effect, i.e., the fluctuation in the price of housing in the urban area of Xuzhou had a positive effect on housing prices in the surrounding areas, and the increase (or decline) in housing prices affected the increases (or decreases) in house prices in other neighborhoods. In terms of the Moran index value, from June 2015 through June 2017, both the global Moran index and the local Moran index were below 0.10, indicating that the overall (local) spatial effect of housing prices in the

urban area of Xuzhou was relatively small. In June 2018, the Moran index rose, indicating that the spatial effect of house prices tended to be stronger.

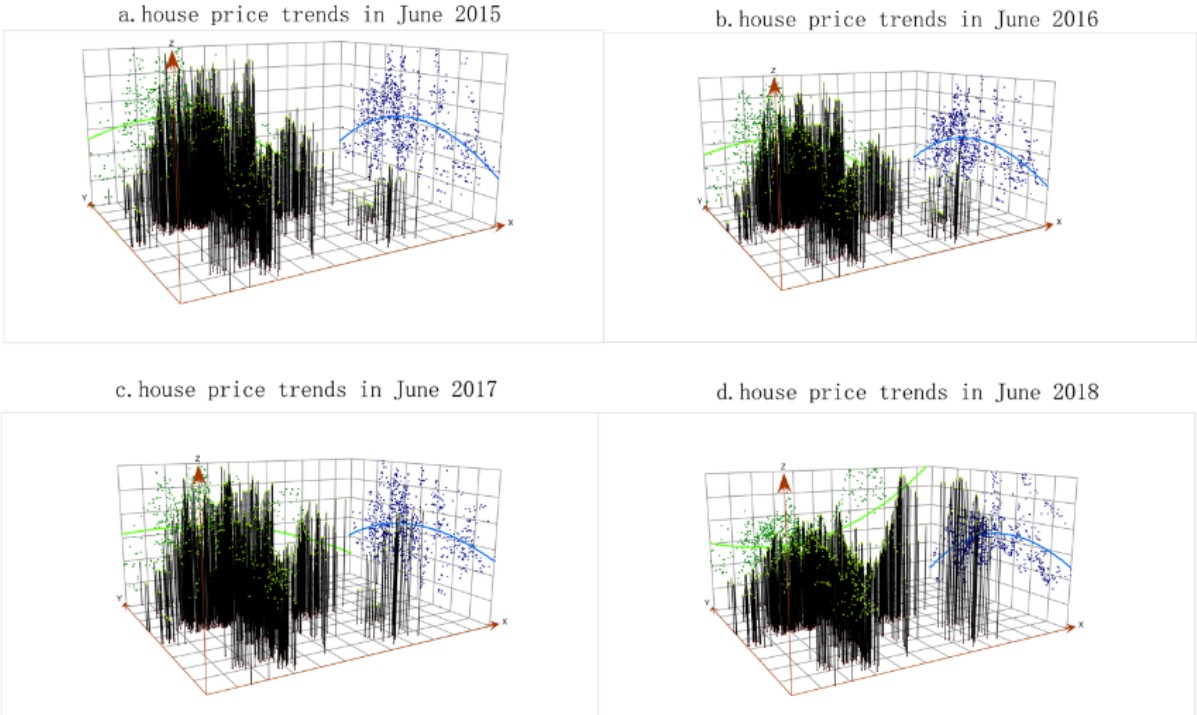

**Figure 5.** Evolution of the spatial pattern of house prices in the Study Area.

**Table 2.** Test table of spatial correlation of housing prices in study area.

| Date | Global Moran Index | | | Local Moran Index | | |
|---|---|---|---|---|---|---|
| | Moran' I | *p*-Value | z-Value | Moran' I | *p*-Value | z-Value |
| June 2015 | 0.074 | 0.001 | 4.190 | 0.074 | 0.001 | 4.320 |
| September 2015 | 0.076 | 0.001 | 4.310 | 0.076 | 0.001 | 4.310 |
| December 2015 | 0.075 | 0.001 | 4.320 | 0.075 | 0.001 | 4.320 |
| March 2016 | 0.078 | 0.001 | 4.420 | 0.078 | 0.001 | 4.420 |
| June 2016 | 0.071 | 0.001 | 3.980 | 0.071 | 0.001 | 3.980 |
| September 2016 | 0.082 | 0.001 | 4.520 | 0.082 | 0.001 | 4.520 |
| December 2016 | 0.077 | 0.001 | 4.340 | 0.077 | 0.001 | 4.344 |
| March 2017 | 0.067 | 0.001 | 3.760 | 0.067 | 0.001 | 3.761 |
| June 2017 | 0.053 | 0.002 | 3.000 | 0.053 | 0.002 | 3.002 |
| September 2017 | 0.122 | 0.001 | 6.630 | 0.122 | 0.001 | 6.631 |
| December 2017 | 0.247 | 0.001 | 13.270 | 0.247 | 0.001 | 13.273 |
| March 2018 | 0.276 | 0.001 | 14.990 | 0.276 | 0.001 | 14.991 |
| June 2018 | 0.289 | 0.001 | 15.400 | 0.290 | 0.001 | 15.404 |

Based on the global and local autocorrelation tests, this study determined the clustering results of housing prices in the study area (Figure 6). It can be seen from the figure that the housing price distribution mainly had four forms: "high-high", "high-low", "low-high", and "low-low". The northwest direction gradually showed a trend of low and low concentration, while the southern and eastern regions did not change much. Because there were many coal enterprises and other industrial enterprises in the surrounding area, the residential area located in the northwest of the research area was affected by the sewage discharge of these enterprises in the early stage of urban development, and the livable experience was poor. There is a saying in the market regarding "half coal and half city dust". In recent years, Xuzhou City has vigorously implemented industrial, urban, and ecological

transformation, paid attention to the excavation of industrial structure, development mode, and economic connotation, and put forward the concept of "coordinated mining and land" to achieve a win-win situation of ecological and economic benefits. The city realized the goal of "The transformation of a city with green mountains and half a city with a lake" and won the China Habitat Environment Award. In addition, because of the relocation of the Xuzhou Municipal Government to the new urban area and the fact that the new urban area was the main direction of urban land expansion in Xuzhou in recent years, residential prices around the government also showed a trend of high-to-high concentration.

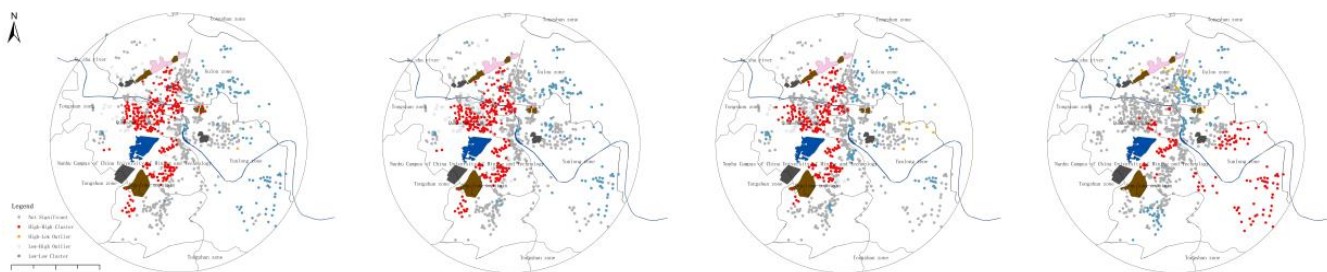

**Figure 6.** Local spatial autocorrelation aggregation diagram of housing price from June 2015 to June 2018.

## 5. Analysis of Factors Influencing Residential Prices

### 5.1. Model Test

Variables were algorithmized to avoid the effect of heteroscedasticity in the regression analysis, and the GTWR model was estimated using MATLAB 2016b software with the Gaussian decay function chosen; the results of the diagnostics are given in Table 3. The spatial-to-temporal distance ratio of the major parameters in the GTWR model was 0.71 and the maximum trace of the matrix was 274.54. The $R^2$ of the model fit was 0.80, indicating a good degree of model fit. To further investigate regional differences in the role of various factors on housing prices, a visual representation of the GTWR results was created based on the GIS spatial analysis, and the study findings were presented in relation to three aspects: building characteristics, location characteristics, and neighborhood characteristics.

**Table 3.** The Statistical Diagnosis of the GTWR Model.

| Indicators | Test Values |
| --- | --- |
| $R^2$ | 0.80 |
| $R^2$ adjusted | 0.80 |
| AIC | 2049.21 |
| Trace of SMatrix | 274.54 |
| Spatial-temporal distance ratio | 0.71 |

### 5.2. Analysis of Results

#### 5.2.1. Analysis of the Architectural Characteristics

As can be seen in Figure 7a, the regression coefficient of age on urban residential prices was largely negative, indicating that overall, the higher the age of the house, the lower its price, suggesting that the increased cost of repairing houses through depreciation dampened house price inflation. From the spatial distribution of the coefficients, the negative effect of house age on house prices was most obvious in Gulou District and Yunlong District, but it showed a weak positive effect around Yunlong Lake in Tongshan District and Quanshan District, indicating that the effect of house age was small in Tongshan District and Quanshan District, and properties in this area showed a gradual increase in value as the age of the house increased. Contrary to previous research findings, for resource-based cities, new houses were worth more than old ones [55], as Tongshan District and Quanshan District were developed later and the housing stock was newer, while the old

city was mostly an old district with older housing stock and higher depreciation. As can be seen from Figure 7b, the overall effect of greening rate on house prices was positive, indicating that the higher the greening rate, the higher the house prices, but the coefficient of the effect of greening rate on house prices had large regional differences. For example, in the neighborhood of the Xu Yunxin River in Gulou District, the positive effect of greening rate on house prices was small, but the positive effect of greening rate on house prices in Yunlong District was relatively large. As can be seen from Figure 7c, floor area ratio had a negative correlation with house prices, indicating that the higher the floor area ratio, the lower the house prices. The Xuzhou City Development Plan (2007–2020) considered the Jiulishan area to be one of the "six clusters" of the city and clearly stated that the development of the area would be intensified, thus weakening the negative effect of plot ratio on house prices to a certain extent.

As can be seen from Figure 7d, the effect of property fees on residential prices was facilitated in the central and newer urban areas and inhibited in the industrial areas in the northeast. This was related to the generally lower property fees in the neighborhoods of the industrial areas and the improved property management in the newer developments. As can be seen in Figure 7e, the impact of dwelling type on regional residential prices was broadly characterized by opposite effects in the north–south direction, with the Yellow River old riverway as the boundary. Areas north of the Yellow River old riverway are mostly industrial areas, with fewer low-rise and mixed housing types, demonstrating the positive impact of housing type on residential prices. The area south of the Yellow River old riverway is close to the city central, where commercial activity was relatively prosperous and real estate development activity was intensive, resulting in an abundance of residential types, thus having a dampening effect on residential prices.

### 5.2.2. Analysis of Location Characteristics

As can be seen from Figure 7f, overall, the distance to planned metro stations was positively correlated with house prices, indicating that the planned metro did not effectively contribute to house price improvements. By region, the impact of planned metro station distance on house prices in key development areas in the city plan was negative, such as in the Jiuli area and the new city, because the planned metro in key development areas facilitated travel for residents, resulting in higher unit prices for housing transactions. The effect on house prices in nonpriority development areas was positive, such as in the main city of the metropolitan area and the old industrial area, because the main city of the metropolitan area is well served by public transport, which, to some extent, weakened the effect of the planned metro station distance on house prices. Workers in the vicinity of the industrial area do not need to travel much to and from the central city and are minimally affected by the distance to the planned metro stations. As can be seen in Figure 7g, there was a negative relationship between the distance from the main road and the house prices of the neighborhoods around Yunlong Lake, indicating that house prices were higher near the main road in the area. Areas such as the new town, south of Yunlong Mountain, the Han Cultural Scenic Area, and the Industrial Zone showed a positive relationship, indicating that prices were lower near the main roads in the area.

As can be seen in Figure 7h, government distance was negatively correlated with house prices. By region, the factor was positively correlated with house prices in the old town and northeast industrial development areas. In the vicinity of Yunlong Lake in the Quanshan District, this factor was negatively correlated with house prices. The neighborhood around Yunlong Lake has beautiful views and is close to the government, and residential prices are higher. Residential prices are least influenced by the main city of the metropolitan area because of its well-developed infrastructure. The industrial area, as a gathering place for old industrial enterprises in Xuzhou, had an incipient scale effect, which, to some extent, weakened the influence of government distance on house prices.

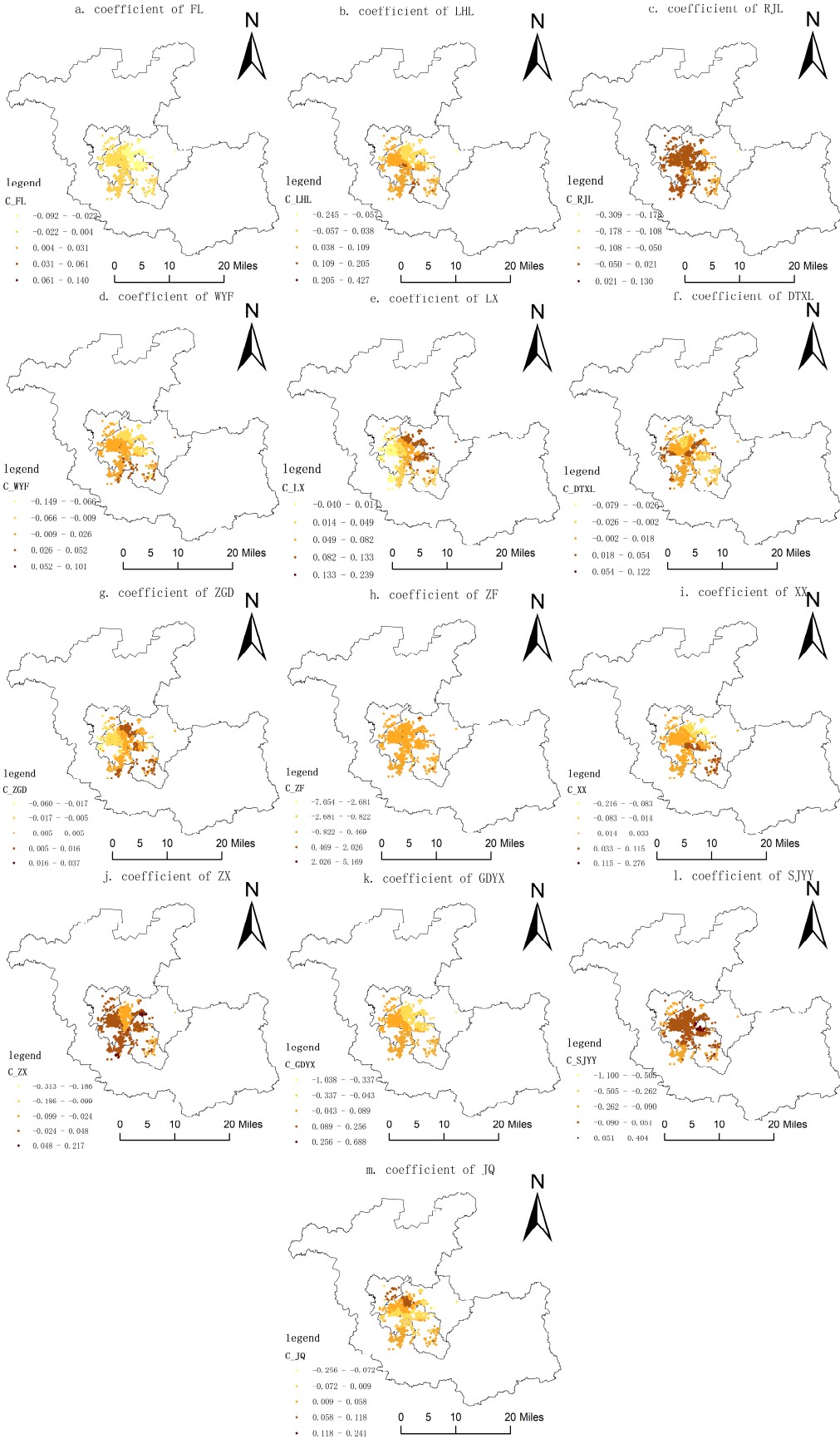

**Figure 7.** The Distribution of Coefficients of Building Effect Factors on the Housing Price.

### 5.2.3. Analysis of Neighborhood Characteristics

As can be seen in Figure 7i, overall, the distance to primary schools showed a negative correlation with house prices, indicating that for most areas, the closer to primary schools, the higher the house prices. The only positive coefficient for this factor was for neighborhoods near new urban areas. It is possible that the impact of primary schools on house prices was weakened by the convenient access to new urban areas. As can be seen from Figure 7j, secondary school distance had an overall negative correlation with house prices. The main reason for this is that secondary schools, as a scarce educational resource, have an elevating effect on the price of surrounding residential properties. In particular, in some districts located in the old industrial areas in the north, the secondary school distance factor had a significant negative effect on residential prices. The number of secondary schools in these areas is sparse, while, on the contrary, in Quanshan District and Gulou District, where there are more secondary schools, the secondary school distance factor had a small or even positive effect on residential prices, indicating that the distribution of educational resources should be taken seriously in the next planning process. As can be seen in Figure 7k, for the northeast region, distance to tertiary institutions had a negative impact on house prices, and for the construction of the new city and the main city of the metropolitan area, distance to tertiary institutions was positively related to residential prices. The possible reasons for this were the low distribution of higher education institutions around the industrial area in the northeast, the high distribution and infrastructure of higher education institutions in the new urban area and the main city of the metropolitan area, the well-developed transport network, and the large number of mobile people in the urban area, which weakened the radiation effect of higher education institutions.

As can be seen in Figure 7l, distance to tertiary hospitals was negatively correlated with house prices overall, a finding consistent with Cao's study [38], where tertiary hospitals have gradually become a scarce resource in cities because of increased health awareness among residents, thereby driving up house prices. In the vicinity of Gulou Xiangshan, there was a positive correlation between tertiary hospitals and house prices, as the distribution of tertiary hospitals was too dense, and the waste generated by hospitals also had a dampening effect on house prices in the vicinity. This finding suggested that there is a range of house price suppression by tertiary hospitals, which was estimated to be from 1.74 to 13.13 km using the functional approximation method. As can be seen from Figure 7m, the positive correlation between distance to tourist attractions and house prices was distributed around Bawang Mountain on North Third Ring Road in Gulou District, indicating that the further away from the tourist attractions, the higher the price of the dwellings. A survey of the area revealed that this phenomenon was because residents in the vicinity of traditional industrial areas do not attach much importance to the impact of tourist attractions, and with the gradual increase in infrastructure in the vicinity, the impact of tourist attractions on property prices is mitigated by the satisfaction of basic leisure and entertainment for residents.

## 6. Conclusions and Discussion

This work studied the evolution of the spatial and temporal patterns of residential prices and the influencing factors in resource-based cities, exploring the characteristics of the spatial and temporal patterns of their house prices based on a dynamic perspective of time series and space. Based on this, a GTWR model was constructed to analyze the influencing factors of residential prices, and the main findings are as follows:

(1) Xuzhou's trend in housing prices in its main urban area, as a typical resource-based city, showed clear phases. Transformations and developments in resource-based cities can provide an external environment conducive to the prosperity of the property market; the monetization policy factor of slum renovation helps promote prosperity in the real estate market, and means of administrative regulation and control can effectively curb excessive house price increases.

(2) The main city of Xuzhou showed obvious spatial divergence in house prices, with urban development planning prompting the peak house price area to gradually

shift to the southeast. The traditional business district and the main city of the metropolitan area were the initial core areas for house prices from June 2015 to September 2017. In September 2017, the construction of new urban areas continued to increase house prices, and they gradually became emerging peak areas of house prices within the main city of Xuzhou. The spatial pattern of Xuzhou's house prices within the traditional business district, the main city of the metropolitan area, and the construction of the new city as the core area of house prices is gradually being set.

(3) In terms of building characteristics, the greatest influence on house prices was the type of dwelling, followed by the green ratio and age of the house, with the remaining factors having a weak influence. There was significant variability in the impact of the functional attributes of urban areas on house prices by residential type, with the impact of residential type on house prices varying between industrial and business areas. In terms of location characteristics, the promotion of house prices by the construction of the metro failed to materialize, and the impact of the distance from the main road and the distance from the government on house prices showed significant regional differences because of factors such as the surrounding attractions and differences in urban functions. The effect of location characteristics on house prices in the well-known scenic area of Yunlong Lake, traditional industrial areas, and key development zones showed mixed results. Neighborhood characteristics had the greatest impact on housing prices, with prices rising near schools in areas where educational resources were scarce, but not near schools in areas where educational resources were abundant. As a scarce urban resource, hospitals at level 3 may contribute to housing prices, but in areas with a high concentration of hospitals, the "neighborhood avoidance effect" may be a deterrent as well.

Based on the results of our study, we found that the price of housing depends, to a large extent, on the distribution of public infrastructure, such as healthcare, education, transportation, and scenic spots in the vicinity, but the specific direction and intensity of the influence of each variable is different. The distribution of public infrastructure in Xuzhou is not sufficiently balanced and regional differences in the impact of public resources such as education and healthcare on urban house prices are relatively pronounced. For example, for most areas, the shorter the distance to the secondary school, the more expensive the house price. On the contrary, in Quanshan District and Gulou District, where the number of secondary schools is higher, the influence of the secondary school distance factor is not significant or even positive. We know from Wen Haizhen's research that educational facilities have a positive capitalizing effect on residential prices with primary and junior secondary schools having a significant school district effect [56]. One of the advantages of the GTWR model is that it allowed us to visualize the spatial and temporal differentiation of each of the influencing factors. At the same time, we know from Wu Bo's research that the GTWR model takes into account not only spatial heterogeneity but also temporal factors compensating for the GWR model's neglect of temporal effects [12]. The GTWR model is effective in analyzing the factors influencing housing prices, and this study can help to rationalize the layout of urban facilities and promote the sustainable development of resource-based cities.

However, there are some limitations to this paper. First, our study used quarterly data from June 2015 to June 2018, which is not a large span of time and thus only yields a three-year period of spatial and temporal house price trends. Second, the quantification of some characteristic variables when dealing with the factors influencing house prices lacks depth. For example, the grade level of the school also plays a role, and factors such as primary and secondary schools and scenic spots can be graded, and future research could deepen these.

**Author Contributions:** Q.Y.: methodology, software, writing—original draft. Y.H.: conceptualization, investigation, supervision. Q.Y.: formal analysis, methodology, visualization, writing—review and editing. Q.Y.: methodology, software, writing—review and editing. All authors have read and agreed to the published version of the manuscript.

**Funding:** The study is supported by the National Natural Science Foundation of China (42171263), Basic Research Funds for Central Universities (140419011).

**Institutional Review Board Statement:** Not applicable.

**Informed Consent Statement:** Not applicable.

**Data Availability Statement:** The datasets used and/or analyzed during the current study are available from the corresponding author on reasonable request.

**Conflicts of Interest:** The authors declare no conflict of interest.

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
