# Peer review of "Spatial–Temporal Variation and Influencing Factors on Housing Prices of Resource-Based City: A Case Study of Xuzhou, China"

_sustainability, doi:10.3390/su15097026_

Round 1

Reviewer 1 Report

Dear Authors,

in my opinion the research is very interesting but the description of your research should be improved. 

1) No clarification of what housing prices are taken into account - housing transaction prices or offer prices?

Make it clear: 261-262: "The housing price data used in the research process comes from the Anjuke website, 261 which is the largest Internet real estate information provider in China"

2) Did You choose prices of flats or detached houses ? I could not find this information in you paper. 

3) Please write it clear: 266-267: "From May 2018 to July 2018, the quarterly data from 266 June 2015 to June 2018 provided by the website is captured ..." I do not understand clearly the time scope of your research. 

4) It is confusing why did You correct the housing price data to the research starting year (2015) according to the consumer price level. 272-273. Why You have not estimated the trend of housing prices? 

5) You wrote about floor area ratio but You have to explain this term and put it in the table 1. 

6) Did You use housing prices as a relation of price to area, or in another way - what way? 

7) There is no description of the limitations of the research carried out

In my opinion, the paper, as amended, is suitable for publication. 

Author Response

1) No clarification of what housing prices are taken into account - housing transaction prices or offer prices?

Due to the confidentiality of the transaction price data, we use the term "offer prices" .We revised the experts' comments as follows:

Since second-hand apartments are considered to be more market-oriented in China and more suitable for reflecting the personal preferences of urban infrastructure [46]. We used the average price of second-hand apartments (in Yuan/m2) of residential communities as the dependent variable.

Make it clear: 261-262: "The housing price data used in the research process comes from the Anjuke website, 261 which is the largest Internet real estate information provider in China"

We have clarified the explanatory variables and elaborated on the shortcomings of the previous study. The specific modifications are as follows:

The housing price data used in the research came from the Anjuke website, which is the largest Internet real estate information provider in China (https://xuzhou.anjuke.com). The housing information provided is relatively comprehensive and reliable, and recent research in the field of real estate has mostly used the housing information provided by it. This study used the web scraping method to collect housing price information.

2) Did You choose prices of flats or detached houses? I could not find this information in your paper.

We chose the general apartment and modified it in the article, specifically:

It should be noted that the price of residential commercial properties as defined herein refers to the average sales price of commercial properties with a 70-year ownership period and the type of housing being "ordinary apartments".

3) Please write it clear: 266-267: "From May 2018 to July 2018, the quarterly data from 266 June 2015 to June 2018 provided by the website is captured ..." I do not understand clearly the time scope of your research.

We have clarified the timing of the study by modifying it as follows:

The study covers the period from June 2015 to June 2018, using quarterly data published on anjuke.com, and a total of 965 residential properties in the study area were collected.

4) It is confusing why did You correct the housing price data to the research starting year (2015) according to the consumer price level. 272-273. Why You have not estimated the trend of housing prices?

To eliminate inflation and make the house price data comparable in the time series, the house price data are revised to 2015 using the Consumer Price Index.

We add an estimate of the time trend in housing prices to the study.

5) You wrote about floor area ratio but You have to explain this term and put it in the table 1.

We have explained the term volume ratio and also placed it in Table 1. The specific modifications are as follows:

The residential floor area ratio refers to the ratio of the total construction area to the total land area, also known as the gross density of the construction area [40].

6) Did You use housing prices as a relation of price to area, or in another way - what way?

Yes, the house price as a relationship between price and area is illustrated in our paper, specifically:

The explanatory variable was the average price of residential commercial properties (in CNY/m2), which aimed to measure the relationship between the price and the size of residential commercial properties.

7) There is no description of the limitations of the research carried out

We have described the study limitations as follows:

However, there are some limitations to this paper. First, our study used quarterly data from June 2015 to June 2018, which is not a large span of time and thus only yields a three-year period of spatial and temporal house price trends. Second, the quantification of some characteristic variables when dealing with the factors influencing house prices lacks depth. For example, the grade level of the school also plays a role, and factors such as primary and secondary schools and scenic spots can be graded, and future research could deepen these.

Reviewer 2 Report

Comments to the authors:

The present study describes the steady development of the real estate market is an important link in the transformation of resource-based cities. Furthermore, it explore that in the process of real estate market development in resource-based cities, the planning department should take into consideration the different functions and division of labour in each region and scientifically formulate urban development plans in order to provide a good external environment for the healthy development of the city's property market.

In many places, in many paragraphs, no citation was observed. The Ms contains large paragraphs, which needs to be breakdown in a small one and relevant. Check the highlighted text to be corrected and improved.

The manuscript has many errors, in many places it is hard to understand the Ms due to long sentences, and there is a scope to improve the quality of the Ms. The conclusion section needs to be reduced. The conclusion section needs to be written precisely.

Specific comments:

L134- Add the citation and check other lines as well for the same mistake.

I have also given the comments in the annotated document, so check it out and address it correctly and sincerely.

Author Response

Responses to Expert Reviews

Modification Description:

L134- Add the citation and check other lines as well for the same mistake.

We have supplemented this section with literature citations and checked the reference citations in the paper.

We have changed the language errors marked by experts and edited the paper in English.

First, we checked the references throughout the text to ensure that the cited ideas were complete and had corresponding literature.

Second, we adjusted the position of Table 1 as suggested by the experts.

Third, we made changes to the English expressions that were not standardized. On this basis, the English editor was asked to assist in checking the language descriptions in the text.

Reviewer 3 Report

The work entitled Spatial-Temporal Variation and Influencing Factors on Housing 2 Prices of Resource-based City: A case Study of Xuzhou, China is an interesting work that shows the factors that influence housing prices. The structure is correct and has all the components of a papar. However, some parts need to be improved. first, both in the introduction and in the theoretical framework, each idea must be correctly cited, there are complete paragraphs with a single quote. It is also important to compare the results with empirical evidence to give more strength to the findings.

Author Response

Comments and Suggestions for Authors

Comments to the authors:

The work entitled Spatial-Temporal Variation and Influencing Factors on Housing 2 Prices of Resource-based City: A case Study of Xuzhou, China is an interesting work that shows the factors that influence housing prices. The structure is correct and has all the components of a papar. However, some parts need to be improved. first, both in the introduction and in the theoretical framework, each idea must be correctly cited, there are complete paragraphs with a single quote. It is also important to compare the results with empirical evidence to give more strength to the findings.

We thank the experts for their comments. We have revised the citations section of the full text to compare the findings with evidence from previous studies. Since there are many changes, we will not list them all here. Changes were tracked in the revised manuscript using red markers.

Round 2

Reviewer 2 Report

Comments

Correct the highlighted parts of the text in the Ms and read the Ms thoroughly  for any error.